# Self-Activating Neural Ensembles for Continual Reinforcement Learning

## Abstract

The ability of an agent to continuously learn new skills without catastrophically forgetting existing knowledge is of critical importance for the development of generally intelligent agents. Most methods devised to address this problem depend heavily on well-defined task boundaries which simplify the problem considerably. Our task-agnostic method, Self-Activating Neural Ensembles (SANE), uses a hierarchical modular architecture designed to avoid catastrophic forgetting without making any such assumptions. At each timestep a path through the SANE tree is activated; during training only activated nodes are updated, ensuring that unused nodes do not undergo catastrophic forgetting. Additionally, new nodes are created as needed, allowing the system to leverage and retain old skills while growing and learning new ones. We demonstrate our approach on MNIST and a set of grid world environments, demonstrating that SANE does not undergo catastrophic forgetting where existing methods do.

## 1 Introduction

Lifelong learning is of critical importance for the field of robotics; an agent that interacts with the world should be able to continuously learn from it, acting intelligently in a wide variety of situations. In marked contrast to this ideal, most standard deep reinforcement learning methods are centered around a single task. First, a task is defined, then a policy is learned to maximize the rewards the agent receives in that setting. If the task is changed, a completely new model is learned, throwing away the previous model and previous interactions. Task specification therefore plays a central role in current end-to-end deep reinforcement learning frameworks. But is task-driven learning scalable?

In contrast, humans do not require concrete task boundaries to be able to effectively learn separate tasks – instead, we perform continual (lifelong) learning. The same model is used to learn new skills, leveraging the lessons of previous skills to learn more efficiently, without forgetting old behaviors. However, when placed into continual learning settings, current deep reinforcement learning approaches do neither: the transfer properties of these systems are negligible and they suffer from catastrophic forgetting (McCloskey & Cohen, 1989; French, 2006).

The core issue of catastrophic forgetting is that a neural network trained on one task starts to forget what it knows when trained on a second task, and this issue only becomes exacerbated as more tasks are added. The problem ultimately stems from training one network end-to-end sequentially; the shared nature of the weights and the backpropagation used to update them mean that later tasks overwrite earlier ones (McCloskey & Cohen, 1989; Ratcliff, 1990).

To handle this, past approaches have attempted a wide variety of ideas: from task-based regularization (Kirkpatrick et al., 2017), to learning different sub-modules for different tasks (Rusu et al., 2016), to dual-system slow/fast learners inspired by the human hippocampus (Schwarz et al., 2018). The core problem of continual learning, which none of these methods address, is that the agent needs to autonomously determine how and when to adapt to changing environments, as it is infeasible for a human to indefinitely provide an agent with task-boundary supervision. Specifically, these approaches rely on the notion of tasks to identify when to spawn new sub-modules, when to freeze weights, when to save parameters, etc. Leaning on task boundaries is unscalable and side-steps the core problem.

There are a few existing task-free methods. Some address the problem by utilizing fixed subdivisions of the input space (Aljundi et al., 2018; Veness et al., 2019), which we believe limits their flexibility.

Recent experience-based approaches such as Rolnick et al. (2018) forgo explicit task IDs, but to do so they must maintain a large replay buffer, which is unscalable with an ever-increasing number of tasks. To our knowledge only one other continual learning method, Neural Dirichlet Process Mixture Models (Lee et al., 2020), adaptively creates new clusters, and we show in our experiments that SANE outperforms it.

Our method approaches the problem head-on by dynamically adapting to changing environments. Our task-agnostic hierarchical method, Self-Activating Neural Ensembles (SANE), depicted in Figure 1, is the core of the method. Every node in the tree is a separate module, none task-specific. At each timestep a single path through the tree is activated to determine the action to take. Only activated nodes are updated, leaving unused modules unchanged and therefore immune to catastrophic forgetting. Critically, our tree is dynamic; new nodes are created when existing nodes are found to be insufficient. This allows for the creation of new paths through the tree for novel scenarios, preventing destructive parameter updates to the other nodes.

SANE provides the following desirable properties for continual reinforcement learning: (a) It mitigates catastrophic forgetting by only updating relevant modules; (b) Because of its task-agnostic nature, unlike previous approaches, it does not require explicit supervision with task IDs; (c) It achieves these targets with bounded resources and computation. In our results, we demonstrate the ability of SANE to learn and retain MNIST digits when presented fully sequentially, a challenging task our baselines struggle with. We also demonstrate SANE on a series of three grid world tasks, showing that SANE works in the reinforcement learning setting.

## 2 MODEL

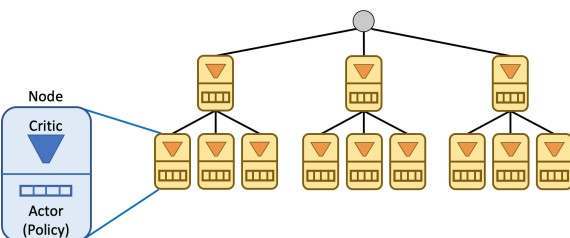

Figure 1: The overall tree structure of the SANE system, here with 2 layers of nodes. Each node contains an actor and a critic.

The structure of our Self-Activating Neural Ensembles (SANE) system is a tree, as shown in Figure 1. Each node in the tree contains both an actor and a critic, described in 2.1. At each timestep we activate a path through the tree from the root to a leaf, at each layer selecting the node whose critic estimates the highest *activation score* (described in 2.2). We then sample an action from that leaf node's policy to execute in the environment. During training, the activated nodes are the only ones updated; all other nodes are unchanged. This enables behaviors selected for by other paths through the tree to remain unchanged.

The tree structure is critical to the success of the system; a node's children can be seen as partitioning the input space that activates the node, specializing their policies to these subspaces. When they differ sufficiently from their parent in performance, they are promoted to a higher level of the tree and obtain children of their own, further partitioning the space. Nodes are constantly being created, promoted to higher levels, or merged as the situation demands (described in Section 2.4).

**Static vs. Dynamic Trees:** One critical question to ask is why use a dynamic structure and not just a huge static tree with only parameter updates? In the case of the static tree, if a path through the network is beneficial for one task, it is likely to be chosen as a starting point for the subsequent task, and updated in-place for this new setting. Examples of this can be found in Jacobs et al. (1991); Shazeer et al. (2017), where specialized losses are necessary to distribute activation across the ensemble of experts, even in the case of a stationary distribution. With SANE, the change of environment would be detected and a new path created, thus minimizing disruption of the previously beneficial behaviors.

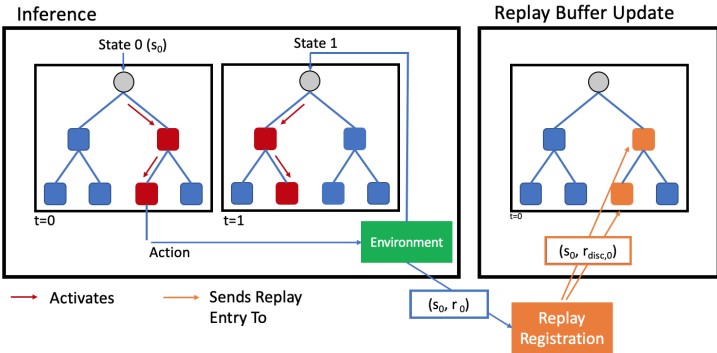

Figure 2: On the left is an example of inference, which results in the generation of actions to execute in the environment. On the right is the associated adding of replay entries to the nodes that activated at $t = 0$, after the collection phase.

## 2.1 NODE CONTENTS

Every node $i$ contains the following: a policy $\boldsymbol{p}_i$, a critic $\mathcal{C}_i$, a replay buffer containing a list of (state $s$, discounted reward $r_{disc}$), and a count of the number of times this node has been used (usage count).

The policy $\boldsymbol{p}_i$ presented here is a vector of $n$ logits, where $n$ is the number of actions available to the agent. The critic $\mathcal{C}_i$ is a neural net mapping a state $s$ to two scalars: the value estimate of the discounted reward received if the current node is activated, $v_i(s)$, and an estimate of the absolute error:

$$u_i(s) \approx |r_{disc} - v_i(s)| \tag{1}$$

With these we can define an optimistic estimate (upper confidence bound) and a pessimistic estimate (lower confidence bound) for what reward a node can achieve:

$$v_{UCB,i}(s) = v_i(s) + \alpha * u_i(s) \tag{2}$$

$$v_{LCB,i}(s) = v_i(s) - \alpha * u_i(s) \tag{3}$$

$\alpha$ is a hyper-parameter representing how wide a margin around the expected value we allow. $v_{UCB}$ is analogous to an upper confidence bound in multi-armed bandit literature: as training proceeds and $v_i$ becomes more accurate, $u_i$ will decrease, providing a natural transition from exploration to exploitation for each node as it learns. This may however act too optimistically in highly stochastic environments, an area of future work.

Each node's replay buffer contains the history of experiences per node and is used to train the node's critic. The usage count is the number of times a node has been activated and is used in the merging of nodes. These are described more in sections 2.3 and 2.4.

## 2.2 INFERENCE

For each observed state $s_t$ from the environment, we activate a path through the tree from the root to a leaf node, whose policy is used to determine the next action to take in the environment, $a_t$. The environment then responds with a reward $r_t$ and a new observation $s_{t+1}$ (See Figure 2).

We find the path recursively: from an activated node, the next node to activate is selected from its children by taking the node with the highest optimistic upper bound of reward. This process continues until a leaf node is reached. After the set of activated nodes is determined, an action is selected stochastically from the activated leaf's policy, and executed in the environment. New nodes are not created mid-episode to keep the tree idempotent, which allows us to readily run environments in parallel.

During inference, three things happen at each timestep to support updating the tree. All active nodes:

- are registered to receive the replay entry for that timestep after the episode completes.
- compare their critic's estimate to their parent's. If they differ sufficiently (Section 2.4), the child will be promoted during the next tree structure update.
- increment their usage count.

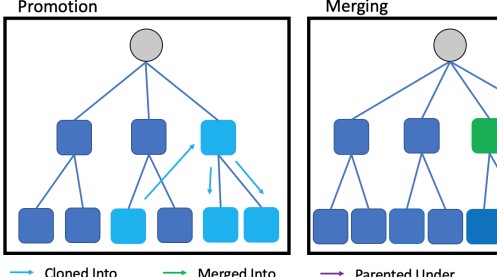

Figure 3: The two parts of updating the structure of the tree: promoting nodes to a higher level, via cloning, and the merging of nodes together when the maximum size has been exceeded.

## 2.3 PARAMETER UPDATES

SANE training updates consist of two steps: parameter updates (replay buffer, policy, and critic) and tree structure updates (creating new nodes, merging nodes, etc.). After $T$ steps of data collection have occurred, we trigger both a parameter update and a tree structure update.

**Replay Buffer Update:** Each node's replay buffer contains the history of the times that node was activated, and is used to train the node's critic. Each entry in the replay buffer contains an input state $s$ and a discounted reward received $r_{disc}$, which is also used in the training of the policies. The discounted reward is computed in the standard fashion, as in Espeholt et al. (2018).

When the addition of new entries results in the replay buffer exceeding its max length, instead of removing the oldest entries, random entries are removed instead. This is to mitigate each critic from developing a recency bias.

**Policy Update:** The policy for each node is trained on-line using REINFORCE (Williams, 1992) with Monte Carlo returns.

**Training the Critics:** The critic for each node is trained off-line using the replay buffer. More specifically, we split a node's replay buffer into batches each of size $k$. The critic has two functions to estimate: $v(s)$ (the expected value) and $u(s)$ (the uncertainty as defined in equation 1). We use L2 loss for both reward and error prediction. Note that this is the slowest step in the training of the SANE tree. Since the nodes all use separate critics, this step can be effectively parallelized. We only train a critic when its node was activated at least once since the last time nodes were trained.

## 2.4 STRUCTURE UPDATES

One of the critical properties of SANE is that the tree structure updates dynamically. Promoting nodes enables existing behaviors to stay unchanged, and merging nodes allows us to keep memory and computation bounded, both depicted in Figure 3. The term promotion is used to refer to cloning a node and placing it in the tree one level up from the original. Merging is the process of combining nodes and is necessary to cap the computational and memory costs that scale with tree size.

**Node Promotion:** Drift is the term for when an environment is non-stationary, e.g. when the reward distribution or the state transition distribution are changing over time. Often where drift occurs, catastrophic forgetting follows because networks update to the new setting, forgetting the old.

To recognize drift, non-leaf nodes at the time of creation have their critic cloned and frozen, creating a static critic called the *anchor*. We compare the prediction of a node's critic to the prediction its parent's anchor makes. We say that sufficient change has occurred when the reward bounds predicted by a node's critic do not include the value predicted by its parent's anchor, which functions as a static baseline. In such a scenario, the node should be promoted up a level of the tree, because we wish to capture the new situation being observed, and minimize disruption of previously learned behavior.

Concretely, we say that sufficient change has occurred when either of the following inequalities hold, for a given input state $s$, node $n$, and parent's anchor $a$: $v_{UCB,n}(s) < v_a(s)$, $v_{LCB,n}(s) > v_a(s)$. In order to avoid an extraneous number of promotions for outliers, a leaf node must have been used more than $k_m$ times since it was created or last promoted.

An implicit assumption here is that the change in reward received is sufficient for distinguishing relevant changes in setting. We believe this to be a general assumption; if, as the task changes, a node's behavior results in the same received reward (and exploration does not expose a better path), then the change was not relevant for determining the optimal policy, and can be effectively ignored.

**Merging Nodes:** To prevent unlimited memory consumption and increasingly long activation times, we limit the number of children each node can have. The limit is defined per level of the tree: level k has limit $\ell_k$. We start by finding the two entries in the set that are closest in policy space. Conceptually, this is so we can find general patterns in the state space that elicit the same optimal policy. To do this we convert the policies from logits to probabilities, then find the two policies that have the smallest L2 distance. L2 was selected because it is symmetric and can be computed batch-wise rapidly. The usage counts, policies, and replay buffers are combined by doing a weighted average of the nodes' contents by usage count, explained in more detail in section A.2. The critic for the new node is trained using the merged replay buffer, after being initialized from the critic of the selected node with higher usage.

Non-leaf nodes additionally update the anchor by combining the nodes' anchors' replay buffers and retraining once. Merging nodes is the exception to the general rule of never updating the anchors.

**Initialization:** The first node in the tree is initialized as a child of the root and has a random critic and a uniform policy. That node is cloned twice, which become the first node's children. This continues until the tree has reached the pre-specified maximum height. From now on, the height of the tree remains fixed. To maintain this, whenever a node is promoted, that node is recursively cloned to generate children until the maximum height is re-achieved.

## 3 EXPERIMENTS

We present two types of experiments that demonstrate the behavior of SANE. The first trains on the digits of MNIST sequentially, to show recall under complete domain shift. The second learns a series of grid world environments, to show that the method works in a reinforcement learning setting. Both never re-train on a task they've seen before; this is to ensure there are concrete points of domain shift. The environments are simple enough that this is not ensured by randomness alone.

In these experiments we only consider two-layer SANE trees, with their sizes ($[\ell_0, \ell_1]$) indicating respectively the maximum number of nodes at the first layer, and the maximum number of children each of those may have. A [20, 4] tree therefore has 20 top layer nodes and 80 second layer nodes, for a total of 100 nodes. The experiments are all run over 4 seeds unless otherwise noted, and the error bars represent the standard error of the mean.

**Baselines:** In both cases we use IMPALA (Espeholt et al., 2018) as the non-continual learning baseline, to demonstrate the effects of catastrophic forgetting, and CLEAR (Rolnick et al., 2018) as a continual learning baseline. CLEAR is a variant of IMPALA that adds an extensive replay buffer to maintain experiences from old tasks. In particular, we selected CLEAR because it is one of the few other methods that does not require task IDs as input. We chose for CLEAR the same number of entries in its replay buffer as the upper bound of the number SANE uses, across all nodes: 409,600 frames. All network and implementation details are given in the Appendix. For the MNIST experiment we additionally use a Neural Dirichlet Process Mixture Model baseline (Lee et al., 2020), described further below.

CLEAR has an implicit trade-off between how easily it learns a new task, and how much it recalls old ones, as defined by the ratio of entries in each training batch that are selected from the replay buffer, versus newly acquired entries. Additionally, if we naively run CLEAR for the same number of timesteps as SANE, we might end up overfitting. We thus additionally consider two types of modification to CLEAR: changing the replay ratio and early stopping. In the latter case we continue to evaluate the network for the rest of the allotted time, to maintain alignment.

**Environments/Datasets:** We conduct two experiments: MNIST and MiniGrid. MNIST allows us to test network behavior in the presence of huge domain shifts as well as a relatively large number of tasks, to highlight how approaches like CLEAR struggle in these scenarios. MiniGrid allows us to demonstrate how SANE avoids forgetting in the reinforcement learning setting. Note that while previous efforts have used Atari, we do not use Atari for a couple of reasons: (a) this setting is particularly ill-suited for continual learning per the desiderata given in (Farquhar & Gal, 2018), for

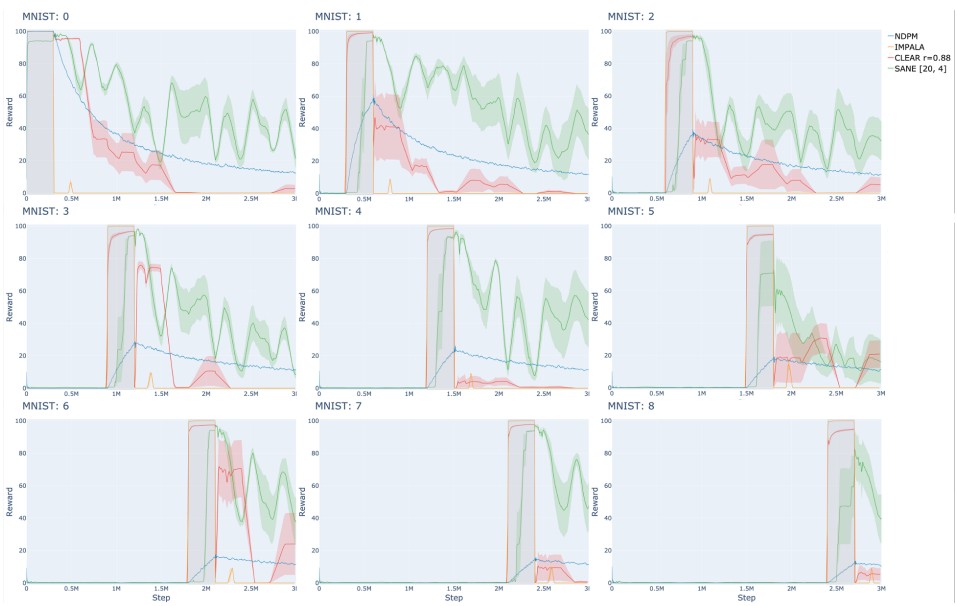

Figure 4: Sequential MNIST results: Each graph visualizes all networks on the given digit. The x-axis is consistent across graphs. Training on a newly introduced digit occurs in the shaded regions, and evaluation to test recall happens everywhere else. SANE is the only method to not catastrophically forget. Digit 9 can be found in the Appendix.

instance as there is little cross-task resemblance, task IDs can be readily inferred (b) the required compute is high (a bottleneck in new model development).

## 3.1 MNIST EXPERIMENTS

MNIST (LeCun et al., 2010) is a dataset of the digits 0-9, each represented by a grayscale 28x28 image. Demonstrating continual learning with MNIST has an extensive history. Permuted MNIST (Kirkpatrick et al., 2017; Schwarz et al., 2018; Goodfellow et al., 2015) trains on the full set of digits concurrently, adding significant pixel-level shifts to change the appearance between tasks. Split MNIST (Farquhar & Gal, 2018) splits the 10 digits into 5 sequential tasks of 2 digits each. Both have assumptions we can now relax; in the former case, the assumption of balanced classes, and in the latter the existence of negative examples for each class being trained. We therefore introduce Sequential MNIST, an alternative task in which digits are trained completely sequentially: first the 0s, then the 1s, etc.

To convert this classification task to an RL task, we define the action to be the selection of a class (i.e. the action space is of size 10), and the reward is defined to be 1 if the correct class is selected, 0 otherwise. We train for 300,000 frames per digit. No normalization is done other than scaling pixel intensities to between [0, 1].

For this experiment we additionally run a Neural Dirichlet Process Mixture Model (Lee et al., 2020) as a baseline, using the code provided by the authors. This is a classification-only method, so for this baseline we do not do the conversion to an RL task described above. We use the default MNIST configuration provided in their code. Due to time constraints NDPM is shown with only one run, though preliminary experiments indicated consistent results. Additionally, we experimented with different configurations of CLEAR by tuning the replay ratio parameter and by early stopping (see Appendix A.7), and selected a replay fraction of 0.875 for our CLEAR baseline.

The results are presented in Figure 4. The networks are trained on the given digit during the timesteps indicated by the shaded region. In the non-shaded regions of the plots, the networks are evaluated on that digit alone, to analyze recall. As this figure visualizes recall and not generalization, these evaluation steps are done on the MNIST Train dataset.

Additionally we test the methods by evaluating them on the Test dataset for the digits cumulative to the most recent digit seen during training. These results are presented in Table 1. "Mem Last" is

| Method | 0 | 0-1 | 0-2 | 0-3 | 0-4 | 0-5 | 0-6 | 0-7 | 0-8 | 0-9 |
|---|---|---|---|---|---|---|---|---|---|---|
| NDPM | **100.0** | 65.0 | 41.0 | 34.0 | 32.0 | 15.0 | 11.0 | 10.0 | 13.0 | 15.0 |
| IMPALA | **100.0** | 53.2 | 32.1 | 24.6 | 19.0 | 15.1 | 14.3 | 13.1 | 10.6 | 10.0 |
| CLEAR (r=0.88) | **100.0** | **94.9** | 54.0 | 39.7 | 36.5 | 19.3 | 20.2 | 22.5 | 11.8 | 15.4 |
| SANE | 97.5 | 82.2 | **75.0** | **57.3** | **47.1** | **50.6** | **29.9** | **26.8** | **28.7** | **33.0** |
| Mem Last | 100 | 50 | 33.3 | 25 | 20 | 16.7 | 14.3 | 12.5 | 11.1 | 10 |

Table 1: The scores of the methods on the MNIST Test dataset, on all digits trained so far, e.g. column "0-3" occurs after training on digit 3 and tests cumulative recall on all four digits to that point. "Mem Last" is the score achieved if the answer given is always the last number seen.

provided as a baseline for what the expected results would be if the network always identifies the last-seen digit.

Only SANE consistently recalls digits after their training period, avoiding experiencing catastrophic forgetting. This setting is particularly challenging for typical methods, because in each training regime there is always only one correct action to take, and it is easy to overfit to providing the singular desired answer. It is worth noting that SANE did not perform flawlessly; there are a few digits (5, 8, 9) that SANE learns inconsistently in the allotted number of frames.

There are a few specific behaviors to note: (a) CLEAR's added replay buffer does provide benefit for recall of earlier digits, though the benefit tapers off as more digits are observed. (b) SANE's curves are reliably cyclic: when SANE begins to do well on a new digit, its performance on prior digits decreases. However, when the next new digit is presented, the performance on prior digits increases again. Our hypothesis is that this is due to overfitting on the current digit, which is mitigated by the negative examples when training on the next digit, leading to some reverse transfer. (c) SANE's performance in the Test condition plateaus lower than the other methods; this is due to the application of a $tanh$ to the logits of the policy to improve stability, meaning even in Test every action has a chance of selection, described further in section A.2. (d) NDPM learns each digit less well than the previous, with only digit 0 actually being mastered, and even that digit still experiences more forgetting than SANE.

## 3.2 MiniGrid Experiments

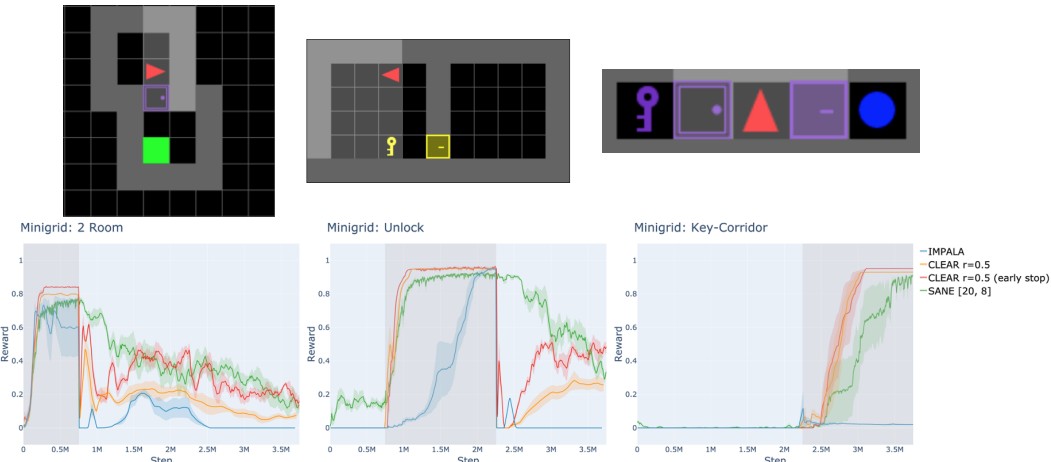

Figure 5: Results for the MiniGrid experiments, in the order in which they were trained. Training occurs in the shaded region, and the rest of the graph depicts recall while the other environments are trained.

MiniGrid (Chevalier-Boisvert et al., 2018) is a lightweight grid world library designed to procedurally generate environments. We train for 750,000 frames on the 2 Room environment, followed by 1.5 million frames on the Unlock environment, and finally 1.5 million frames on the Key-Corridor environment, depicted in Figure 5. Each is randomized at the completion of each episode. The observation is an egocentric representation of the environment, capturing state information about the 7x7 square in front of the agent.

The training results are presented in Figure 5. As with MNIST, the shaded region indicates when that environment was being trained, while the unshaded parts are evaluation steps on that environment,

to visualize recall. We experimented with different configurations of CLEAR (see Appendix A.7), and for MiniGrid selected early stopping, though a replay ratio of 0.75 performed comparably.

SANE's performance is comparable to that of CLEAR with early stopping, indicating that SANE can successfully mitigate catastrophic forgetting in RL environments, even improving on performance in places. CLEAR without early stopping performs worse than both, indicating that it is likely overfitting. The fact that SANE trains extensively on the Unlock environment after plateauing (for approximately an additional 1 million timesteps), may indicate that SANE is less susceptible to overfitting, potentially another benefit of the modularity of our architecture. This benefit is not to be understated; without inherent resilience to overfitting, agents that are continuously acting in the world must have some automatic mechanism to detect it, a feature which most CL methods do not presently have, with the exception of (Aljundi et al., 2018). IMPALA exhibits catastrophic forgetting as expected, even though it struggles with learning the second two environments.

## 4 ANALYSIS OF SANE

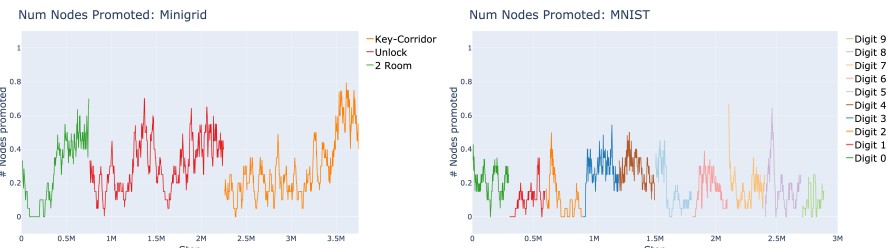

Figure 6: The nodes promoted by SANE while running the MiniGrid (left) and MNIST (right) experiments. For clarity they have been smoothed by averaging over a window of size 20.

**Creation of new nodes:** Figure 6 shows when SANE decided to promote nodes during one run of each experiment. For MiniGrid, it steadily increases per task, whereas for MNIST the pattern is less distinct. The former behavior is expected; low as negative examples are provided to the current best nodes, then high as nodes improve over their parents. It may remain high because the tree starts to run out of capacity around 300k steps, and promotion+merging nodes is less sample efficient than promotion alone. The behavior of MNIST is less clear though, and future work will investigate it more closely. We provide some further intuition for the behavior of the SANE network by visualizing a few states that activate the same node in Figure 7, located in the Appendix.

**Parameter and architecture analysis:** Unfortunately SANE is relatively slow (about a day to train Minigrid, versus an hour for CLEAR) and considerably compute-intensive, so a complete grid search and set of ablations has been out of reach. Informal experimentation has shown little sensitivity to most parameters, with the primary exceptions being the size of the nodes and the number of epochs they are trained for, as well as $\epsilon$ in $\epsilon$-greedy. The former two are particularly inconvenient because they are highly tied to the runtime of the method.

## 5 RELATED WORK

The goal of continual learning, or lifelong learning, is to create an agent that can learn to accomplish a wide variety of tasks sequentially (Lesort et al., 2019). To accomplish this a system must balance stability (the extent to which existing knowledge is retained) and plasticity (how readily new knowledge is acquired) (Parisi et al., 2018).

The former in particular has posed a substantial problem for neural networks, which undergo catastrophic forgetting, tending to completely forget old tasks when presented with new ones, often quite abruptly (French, 2006). Attempts to correct for this tend to fall into these broad categories, which we discuss below: dynamic networks, dual memory, regularization, rehearsal, sparse-coding, and ensembles (Parisi et al., 2018; Kemker et al., 2017).

**Dynamic architectures and dual memory:** One example of dynamic architectures, where resources are added as-needed, is CHILD (Ring, 1997), which determines when to create new units by analyzing average errors observed in network weights. Another dynamic architecture method is Progressive Networks (Rusu et al., 2016), where new neural networks ("columns") are added as new tasks are encountered, with lateral connections created to the networks of previously learned tasks.

Progress and Compress (Schwarz et al., 2018) adapts this into a dual-memory method by maintaining two columns and regularly distilling the active column into the knowledge-base column.

**Regularization:** Elastic Weight Consolidation (Kirkpatrick et al., 2017) and Synaptic Intelligence (Zenke et al., 2017) are examples of regularization. Both automatically learn an importance measure for the individual neuron weights for particular tasks, such that new tasks minimally influence weights that were important for prior tasks. The method presented by Aljundi et al. (2018) is a task-free regularization method that uses features of the loss as it is updated: when a plateau is detected after a peak, the parameter importance weights are updated.

**Rehearsal:** One rehearsal method is CLEAR (Rolnick et al., 2018), which maintains a replay buffer over old tasks, using reservoir sampling and a behavioral cloning loss to ensure the network continues to respond appropriately to the inputs of old tasks, giving this method some regularization as well. CLEAR is built on top of IMPALA (Espeholt et al., 2018), an asynchronous non-continual learning actor-critic method. Another rehearsal method is Maximally Interfered Retrieval (Aljundi et al., 2019), which trains on the entries from its buffer that a proposed update (using recently observed data) most negatively impact. MIR also introduces a variant using a generative model. Both of these do not require task IDs.

**Sparse coding and ensembles:** Fixed Expansion Layer Networks (Coop et al., 2013) (FEL) is an example of both a sparse-coding and an ensemble method, though not one demonstrated in RL environments; it introduces a layer that is sparsely activated, such that the error signal does not propagate through all weights, and creates an ensemble of these to improve results. PathNet (Fernando et al., 2017), which may also be considered an ensemble method (Kemker et al., 2017), uses an evolutionary strategy to select a path through the network. The usage of specific paths instead of the whole network is a similarity PathNet shares with SANE. However, the method still requires task boundaries to determine when to freeze paths, a key difference from SANE.

**Task-boundary free:** In addition to CLEAR, MIR, and Task-Free Continual Learning, there are a few other explicitly task-boundary free works. Goyal et al. (2019) also present a distributed network of primitives that independently determine the score used to activate them. However, their primitives are constrained in terms of what part of the input space they see, and new primitives are not created over time. Gated Linear Networks (Veness et al., 2019) introduces a hierarchical backpropagation-free method that gates neuron selection by associating inputs with predefined context vectors. Continual Neural Dirichlet Process Mixture Models Lee et al. (2020) leverages the ability of standard DPMMs to handle clustering with an arbitrary number of clusters, increasing as necessary when new data is not well-explained by an existing cluster. Finally, (Zeno et al., 2019) approach task-free continual classification by estimating a Bayesian posterior online.

Additionally, some of the components of SANE have occurred in non-CL literature. For instance, Mnih & Hinton (2009) and Shen et al. (2017) utilize a method that selects a leaf node by making choices at each internal node of a tree, though tree creation is by clustering not by drift detection. Ghosh et al. (2017) presents Divide and Conquer, an idea conceptually similar to SANE; it partitions the input space and learns an ensemble of specialized policies, though they cluster with K-Means while SANE forms partitions automatically.

## 6 CONCLUSION

The core challenge behind continual learning is knowing how and when to adapt to novel situations without overwriting previous behavior. Most existing methods side-step the problem by allowing the task boundaries to be specified, giving a clear signal for when to add parameters, freeze them, reinitialize them, etc. Our method, Self-Activating Neural Ensembles, needs no such crutch. It automatically detects and adapts to drift in the environment by adding new nodes to the SANE tree. Additionally, since only the necessary set of nodes is activated at each timestep, nodes that did not participate are not updated with the new interaction data and therefore are not susceptible to catastrophic forgetting. Finally, our method automatically merges nodes as necessary, ensuring the resources consumed remain bounded.

**Future Work:** Improving the speed of training has been a bottleneck, and is one direction for future work. Other directions include experimenting with more layers, sharing parameters between nodes, having a common pre-encoder akin to the visual cortex, actively mining negative examples from other nodes, adapting to continuous and stochastic domains, leveraging distributional models, and many others.

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

# A  APPENDIX

Code is provided in the supplementary.

## A.1  NODE VISUALIZATIONS

Here we see four instances of states the same node was activated. These examples were simply the last four usages we recorded where this node was activated, not cherry picked from a larger set. In all cases we can see that the ideal next action is to move forward, and indeed that is what this node's policy encodes for.

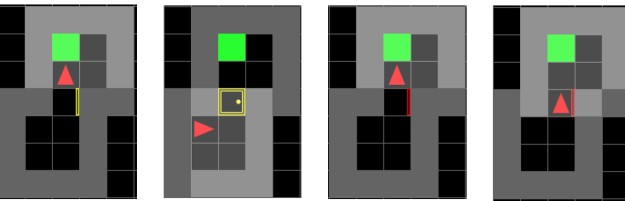

Figure 7: An example of four instances where the same node was triggered. When this node fires, the action taken is for the agent to move forward.

Additionally, in the supplementary there are two videos, minigrid_tree.mp4 and mnist_tree.mp4, which demonstrate how the network grows over time. Nodes are created at the bottom of the video in pink, and darken to blue as they get used, according to the same scaling as usage counts are scaled, see below.

## A.2  SANE IMPLEMENTATION DETAILS

There are a number of details left out of the main SANE explanation, to streamline the explanation down to the most significant concepts. These details are explained here, roughly in decreasing order of significance.

- The entries in the replay buffer are importance-weighted when being used to train the critic. In particular they are scaled by the simple ratio: $\frac{P_{cur}(a)}{P_{old}(a)}$, or in other words the ratio of the probability of the selected action using the node's current policy to that if the node's policy at the time the action was originally taken.

- When merging nodes, only the first 3/4 of the set of nodes is eligible to be merged. This is under the premise that the most recent nodes were likely newly created, and we're giving them a little longer to be used before we collapse it into another node.

- During training SANE operates in an epsilon-greedy manner, selecting a random action some $\epsilon$ fraction of the time, to encourage exploration.

- All nodes' policies are scaled as $3tanh(P)$, meaning each logit falls in the range (-3, 3). This is for a few reasons. (1) This stabilizes the REINFORCE gradient applied to the policy. (2) Even if a node is exceptionally confident in an action, when it encounters a new situation it must still have some non-zero chance of trying another action (related to epsilon-greedy). (3) Raising the probability of a previously-disadvantaged action to a range where it gets selected frequently should not take a large number of examples.

- We maintain 3 sets of usage counts: (1) The total number of times the node has been activated across its entire ancestry (i.e. including the count of the node it was cloned from), roughly representing how many unique replay entries the critic has been exposed to. During merge this count becomes the average of the two nodes'. This is the usage used in weighting policies and replay buffers during merge. (2) The usage count since this node was created or last cloned. Used for determining whether the node is eligible to be cloned again. (3) The usage count since the last time the node was trained, only used to determine whether to train the node's critic during the update step.

- Usage count (1) from the above point is not used directly in determining a node's weight; instead the weight is given by $max(1, tanh(usage\_count/20000) * 100)$. These constants have not been highly tuned, and work surprisingly well regardless of task. Ensuring a minimum of 1 means that under-used nodes are never completely ignored during a merge.

- To adjust for the fact that rewards are not consistently scaled across tasks, we maintain a $max\_reward$ constant, set to the absolute value of an observed reward that exceeds the current $max\_reward$. This value is multiplied by .999 every timestep. All rewards are scaled as $10 * reward/max\_reward$, to scale their values between [-10, 10].

- The critic network's weights are initialized according to a normal distribution with mean 0 and standard deviation $0.001/in\_features$. In other words, the values are very close to 0.

- To adjust for the fact that the very first node's anchor is initialized randomly, and thus not effective as a baseline, we have an edge-case way to initiate the promotion of a node: when it is used more than 10,000 times. (Usage count 2 above.)

- We take a leaf out of IMPALA's book and, since our critics are updated asychronously, every node has a duplicate critic that is what gets trained, and then its weights are copied into the shared critic that is used during data collection. This does mean critic weights change mid-data collection, but not mid-training.

- Every training step we update the non-leaf nodes' policies to be the weighted average of their children's. These policies are used for two purposes: (1) if the node is cloned, its policy will end up as the policy of a leaf, and thus used, and (2) the policies are used to determine which nodes to merge, so a reasonably meaningful one is beneficial. These aspects are primarily important in trees with more than two layers.

- Periodically (every 100 updates) we send to each non-leaf node a batch of entries from each of its children. This is in case one of the children has been getting activated to the exclusion of the others, ensuring the parent still represents its other children.

- When bootstrapping the estimated value for the last entry in the replay buffer to compute the discounted rewards, we found it helpful to multiply that value by 0.9. This is so if the estimate is wrong, it doesn't perpetuate itself via the discounted rewards entered into the replay buffer.

## A.3 SANE Experiment Details

Many of SANE's constants are common between experiments:

| Description | Value |
|---|---|
| Policy learning rate | 1e-4 |
| Critic learning rate | 1e-3 |
| Replay buffer size | 4096 |
| Min usages for promotion $k_m$ | 1500 |

**MiniGrid constants:**

| Description | Value |
|---|---|
| Num parallel envs | 4 |
| Timesteps between updates ($T$) | 1000 |
| Error scale $\alpha$ | 0.5 |
| Merge ratio | 1 |
| Critic train epochs | 1 |
| Replay batch size ($k$) | 4096 |
| Reward discount $\gamma$ | .99 |
| Epsilon $\epsilon$ | 0.02 |

**MNIST constants:**

| Description | Value |
| --- | --- |
| Num parallel envs | 1 |
| Timesteps between updates ($T$) | 250 |
| Error scale $\alpha$ | 1.0 |
| Merge ratio | 3/4 |
| Critic train epochs | 2 |
| Replay batch size ($k$) | 1024 |
| Reward discount $\gamma$ | 0 |
| Epsilon $\epsilon$ | 0.04 |

MNIST gets divided into more batches per training cycle, MNIST's discount factor must be 0 because the images presented to the agent have no correlation to each other in time, and MNIST gets a higher exploration rate.

### A.4    BASELINE IMPLEMENTATION DETAILS

For IMPALA we used the Torchbeast implementation (Küttler et al., 2019). We built CLEAR on top of this base. We used the same constants for both baselines, except IMPALA used an entropy cost of 0.0001.

Code is available in `continual_rl/policies/impala/torchbeast/monobeast.py`. torchbeast's default values are available in `continual_rl/policies/impala/impala_policy_config.py` Listed here are the ones we changed:

**MiniGrid constants**

| Description | Value |
| --- | --- |
| Unroll length ($k$) | 20 |
| Entropy cost | 0.005 |
| Learning rate | 0.006 |
| Num learner threads | 1 |
| Discounting | 0.99 |

**MNIST constants**

| Description | Value |
| --- | --- |
| Unroll length ($k$) | 20 |
| Entropy cost | 0.01 |
| Learning rate | 0.001 |
| Num learner threads | 1 |
| Discounting | 0 |

### A.5    MINIGRID REMAINING RESULT

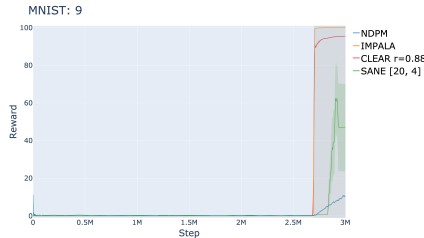

Figure 8: The result of training on digit 9, annexed here for want of space.

## A.6 Network Specifications

To simplify things, we adopt the convention of describing 2d convolutional layers with $Conv(c, w, s)$, indicating there are $c$ output channels, window size $w \times w$, and stride $s$. Similarly, we denote max pooling over a window size of $w \times w$ with $MaxPool(w)$. Finally, we denote a linear layer with output $c$ channels with $Lin(c)$.

In all cases the non-linearity used is ReLU, and for simplicity remains implied.

### A.6.1 Base Networks

For MiniGrid the base encoder, shared by SANE, CLEAR, and IMPALA is

$$Conv(16, 2, 1) \rightarrow MaxPool(2) \rightarrow Conv(32, 2, 1) \rightarrow Conv(64, 2, 1) \rightarrow Flatten \rightarrow Lin(32) \tag{4}$$

For MNIST the base encoder is:

$$Conv(24, 5, 1) \rightarrow MaxPool(2) \rightarrow Conv(48, 5, 1) \rightarrow MaxPool(2) \rightarrow Flatten \rightarrow Lin(32) \tag{5}$$

For CLEAR and IMPALA, the output of these encoders is concatenated with a one-hot of the last action and the last reward, and projected using a linear layer to the final output size: the total number of actions (10 for MNIST, 7 for MiniGrid).

For SANE, the encoder is followed by

$$encoder \rightarrow Lin(128) \rightarrow Lin(128) \rightarrow Lin(2) \tag{6}$$

Where the the 2 output values are the expected value $v$ and the uncertainty $u$ for that node. This gives a total of about 75,000 parameters for MNIST and around 35,000 for MiniGrid.

### A.6.2 Increased Parameter Networks

A number of iterations were tried for these larger networks, particularly with emphasis on maintaining certain bottleneck layer dimensions, though we do not claim to have spanned the whole space of options. These are what were used in the paper:

For MiniGrid, giving around 1.2M parameters:

$$Conv(1024, 2, 1) \rightarrow Conv(1024, 1, 1) \rightarrow MaxPool(2) \rightarrow Conv(32, 2, 1) \rightarrow Conv(64, 2, 1)$$
$$\rightarrow Flatten \rightarrow Lin(32) \tag{7}$$

For MNIST, giving around 5.5M parameters:

$$Conv(24, 5, 1) \rightarrow Conv(2048, 1, 1) \rightarrow Conv(2048, 1, 1) \rightarrow Conv(24, 1, 1) \rightarrow MaxPool(2)$$
$$\rightarrow Conv(48, 5, 1) \rightarrow Conv(1024, 1, 1) \rightarrow Conv(1024, 1, 1) \rightarrow Conv(48, 1, 1) \rightarrow MaxPool(2)$$
$$\rightarrow Flatten \rightarrow Lin(32) \tag{8}$$

## A.7 CLEAR REPLAY FRACTION COMPARISON

### A.7.1 MNIST

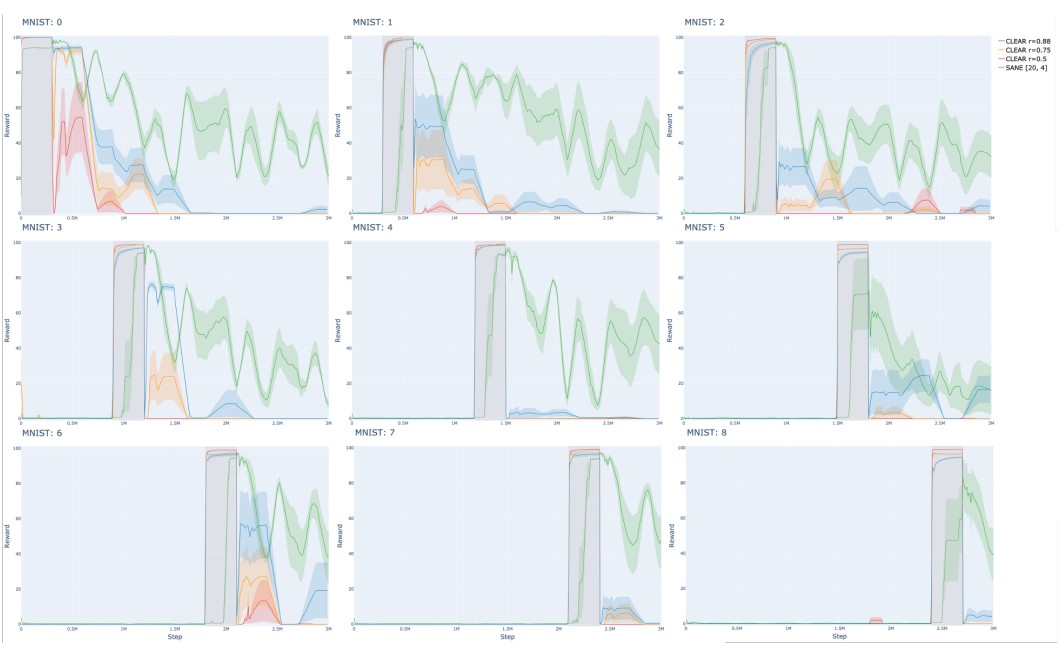

Figure 9: Comparison of CLEAR methods to observe the trade-off between speed of learning a new task and recall of old tasks, in MNIST. The number represents the fraction of the training batch (of size 8) that is from the replay buffer. SANE also included for comparison.

In Figure 9 we present a comparison of CLEAR runs, varying the fraction of each training batch that is sampled from the replay buffer. We can see that the higher ratios do indeed both learn more slowly (for instance digit 2), and remember previous tasks longer. Higher ratios did not reliably learn the digits, so they were omitted from the analysis. From this we see that r=0.88 does the best, so that is what we use in the comparison in Figure 4. We additionally ran early stopping (stopping after exceeding a threshold of 0.97), but it exhibited no recall on past experiments, so we omitted it to reduce clutter in the graph.

### A.7.2 MINIGRID

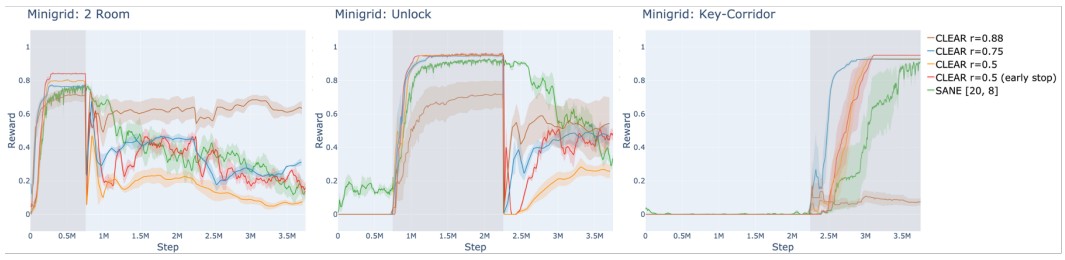

Figure 10: Comparison of CLEAR methods to observe the trade-off between speed of learning a new task and recall of old tasks, in Minigrid. The number represents the fraction of the training batch (of size 8) that is from the replay buffer. SANE also included for comparison.

In Figure 10 we see our comparison for Minigrid, with the same method as used for MNIST. Early stopping is done when the value exceeds (0.78, 0.95, 0.92) respectively, selected by observing prior runs. Note replay ratio of 0.88 performs poorly in the second task and does not succeed at all at the third task, allowing for the superior recall of the first task. Early stopping and a replay ratio of 0.75 perform comparably; we select early stopping for our comparison due to better asymptotic performance in 2 Room. Note that, by chance, every experiment type (including SANE) had one run fail to learn task 3 within the allotted time, and was thus re-run.

