# OpenReview forum: "Self-Activating Neural Ensembles for Continual Reinforcement Learning"
_ICLR.cc/2021/Conference — Reject_

### Official Review · AnonReviewer2 · 2020-10-26
**Interesting idea but lacking in empirical validation and justification of architectural choices**

**Rating:** 5
**Confidence:** 4

**Review:**

Post-rebuttal update:

I thank the authors for their responses to my queries and for updating the paper. I think the paper has improved, the main reason being that the replay-ratio parameter sweeps for CLEAR in the MNIST experiments show that SANE does improve the stability-plasticity tradeoff, in particular the runs with a 0.97 replay ratio (only shown in the rebuttal pdf but I think should make it into the appendix of the paper), which show that the memory of CLEAR cannot be improved to match SANE's without severely affecting its ability to learn new tasks. The only other set of experiments, the sequence of three minigrid tasks, however, do not show an overall improvement over CLEAR, and so overall the experimental evaluation is still weak. It is hard to judge how good the method is without extending the range of baselines (I appreciate the addition of the and task settings. As a result, I am only increasing my score from a 4 to a 5, but I do think SANE is an interesting method and I encourage the authors to keep working at it to prove its effectiveness.

---------------------------
Original review:

This paper presents SANE (Self-Activating Neural Ensembles) a new dynamic architecture method for mitigating catastrophic forgetting in neural networks in the context of reinforcement learning that is task-agnostic. The architecture consists of a tree where each node comprises its own value critic, policy and replay buffer. Action selection at each time step begins by starting at the root of the three and activating the child that predicts the most optimistic value, and repeating this procedure from the activated child until a leaf node is reached, whose policy is used to act with. Training has two components: (i) the parameters in activated nodes are updated using REINFORCE, and (ii) structural updates are made to the tree whereby children that are sufficiently different to their parents are promoted up the tree and given their own children, and children that are similar to each other are merged in order to limit growth of the tree. The method is empirically compared to two baselines, IMPALA and IMPALA with CLEAR (an experience-replay based continual learning method), on two sequences of tasks, sequential MNIST and a sequence of three egocentric gridworld tasks (MiniGrid). On sequential MNIST, SANE displays less forgetting but less adaptivity than the baselines, and on the gridworld tasks the results are mixed.

SANE is an interesting and inventive method but I do not think it is ready for publication yet because (i) the experiments do not sufficiently support the claim that SANE improves catastrophic forgetting over the (limited set of) baselines, and (ii) SANE has a lot of components and hyperparameters but it is not clear which aspects are important for it to work. I believe the paper could be much improved by (i) demonstrating that the baselines have been adequately tuned (and perhaps increasing the set of methods compared to, e.g. with another dynamic architecture method), (ii) including a number of ablation studies or other experiments that better elucidate the inner workings of SANE and justify architectural choices.

Positives:
	•	SANE incorporates a number of interesting mechanisms that might intuitively be useful for mitigating catastrophic forgetting. For example, by only updating activated nodes, modularity is preserved during training in a way that can protect forgetting in other parts of the network. Also, the structural updates provide an interesting way of expanding and contracting the architecture as necessary under fixed resource constraints - an important challenge for continual learning.
	•	The problem that SANE tackles, i.e. task-agnostic continual learning, is an important one for which there are not a large number of approaches in the literature.
	•	An effort is made to equalise the number of parameters used by the baseline agents and SANE in the experiments for a fair comparison, and experiments are repeated with several random seeds.

Concerns:
	•	SANE is claimed to improve on the baselines of IMPALA and CLEAR on the axis of mitigating catastrophic forgetting, but the experiments do not demonstrate this clearly.
	◦	In the sequential MNIST experiments, SANE retains some performance on earlier tasks as training progress but the performance of both IMPALA and CLEAR seems to drop almost instantaneously after the task is switched. On the other hand, both IMPALA and CLEAR learn the new task extremely quickly compared to SANE, which usually does not even reach the same maximum level of performance as the baselines. This suggests that SANE shifts the tradeoff of adaptivity vs. remembering towards remembering, but doesn’t show that the tradeoff is improved vs the baselines. In the CLEAR paper, there is an important hyperparameter that mediates this tradeoff, which is the proportion of replay experiences vs. new experiences used for training - this parameter should be tuned here to be able to show that CLEAR is indeed inferior to SANE on this task. (Judging from the code it seems that a 50/50 split is used, is this correct?)
	◦	In the sequence of three MiniGrid tasks, even though the performance of the CLEAR agent on the first task eventually drops below that of the SANE agent, the SANE agent initially deteriorates a lot faster. For the second task there seems to be no significant difference between agents. Additionally, the SANE agent is much slower to learn on the third task, never reaching the performance of the CLEAR agent, again demonstrating that the SANE agent is trading off adaptivity for slightly better memory. It’s not evident that SANE is improving the tradeoff vs. CLEAR without tuning the latter’s hyperparameters.
	•	It is difficult to discern which elements of SANE are important for its performance in order to justify some of the architectural choices. For example:
	◦	The critics are trained to estimate both the value and the variance of the value, and, in the inference stage, the node with the highest upper bound of estimated reward is chosen. How important is this UCB-like policy for the performance of the agent, and the value of the hyperparameter alpha that scales the standard deviation term? Would SANE work just as well if just the value estimate was used to choose the next node? Could this lead to a wrong solution if applied in a highly stochastic environment?
	◦	The policies of the nodes in SANE use REINFORCE but IMPALA and CLEAR use VMPO; is there a reason for this difference?
	◦	In the structural updates, for node promotion the value functions of a child and its parent are compared, but for node merging, the policies are compared - is there a rationale behind these choices?
	◦	In the MiniGrid experiments, SANE is run with either 10 or 15 top level nodes, resulting in a relatively big difference in performance for the 1st and 3rd tasks, giving some indication of the sensitivity of this hyperparameter. Why were 20 top-level nodes chosen for the MNIST experiments and was the performance sensitive to this choice? How about other hyperparams such as number of child nodes and the number of times a node has to have been used before it can be promoted?

Other comments:
	•	The number of baselines used for comparison is limited, especially given that one of them (IMPALA) is not designed for continual learning. It would at least be good to cite other task-agnostic methods such as [1,2].
	•	It is slightly odd to frame sequential MNIST as an RL problem, when it is a supervised learning task.
	•	A visualisation is given in the appendix showing an example of a node that seems to represent a policy that always moves the agent forward one step in the gridworld. Presumably this is shown to indicate the specialisation of modules in the tree; it would be very interesting to see more evidence of specialisation/diversity across modules since this is one of the motivating factors for the architecture and also to see how the tree structurally adapts at task switches - are there more promotions when the data distribution changes?

[1] Aljundi, Rahaf, Klaas Kelchtermans, and Tinne Tuytelaars. "Task-free continual learning." Proceedings of the IEEE Conference on Computer Vision and Pattern Recognition. 2019.
[2] Zeno, Chen, et al. "Task agnostic continual learning using online variational bayes." arXiv preprint arXiv:1803.10123 (2018).

---

> ### Author Response · Authors · 2020-11-20
> **Response to Reviewer 2 (Part 1)**
>
> Thanks for the thorough review, we really appreciate it.
>
> **Trade-off of adaptivity vs memory:**
>
>
>
> It is correct that experiments were initially run with a ratio of 0.5; the CLEAR paper said that is what they used by default, and their results showed little sensitivity to the parameter. However, as pointed out, there is likely to be a tradeoff between adaptivity and memory and therefore we did several runs varying this parameter. Results can be seen in Figure 5, where a higher number means a larger fraction of the training batch is from the replay buffer. These results were done with a batch size of 8. We can see that the higher the fraction, the less well it trains, but the more it remembers, though the improvement still does not generally come close to SANE.
>
>
>
> With a batch size of 8, the differences in adaptivity can be difficult to differentiate (the maximum ratio being 7/8), so we also ran with batch size 128, which allowed us to look at even higher ratios, as can be seen in Figure 6.
>
>
>
> These latter experiments were run with a lower frequency of CL evaluation for speed. We can again see that at the higher ratios, learning is quite bad, but memory can persist longer. Given that we had to take the ratio up to 0.97 to start seeing CLEAR be comparable to or beat SANE (digits 2 and 5), at which point other digits are failing to learn, I think we can say SANE has a better adaptivity/memory tradeoff.
>
>
>
>
>
>
>
>
>
>
>
> **It is difficult to discern what is important.** This point had several parts, so we've broken them out below. Overall we haven’t done a full analysis for every parameter, due to the slow runtime of SANE.
>
>
>
> **How important is this UCB-like policy for the performance of the agent, and the value of the hyperparameter alpha that scales the standard deviation term? Would SANE work just as well if just the value estimate was used to choose the next node?:** Many iterations of attempting to use only the value estimate were tried. The problem is three-fold. One is that without UCB, there is substantially more initial sensitivity to network initialization; it’s easy for a single node to be hyper-fixated on, and with only epsilon-greedy to get out of that hole, it can be slow. The second is that when moving to a new setting, we are again largely at the whim of epsilon-greedy. Having some metric of “this estimate is wrong, let’s investigate further” speeds up adaptation. The third is that it was by far the easiest and most successful metric we developed for knowing when to create new nodes (via node promotion), and using the extra information since we were already collecting it seemed natural. While we have not yet extensively tested the dependence on alpha, results showed resilience to alpha (paper reports alpha=1 and we get even better result with alpha=0.5).
>
>
>
> **Could this lead to a wrong solution if applied in a highly stochastic environment? :** Say there is an environment where a particular action gives an equal chance of receiving reward -1 or 1. The mean would be 0 and the error 1, so the UCB would be 1. This would be the higher than an action that reliably gives a reward of 0.5, meaning SANE would select the suboptimal action. It seems possible that in stochastic environments, the tradeoff between estimate accuracy and exploration would lean towards estimate accuracy, meaning we should instead select based on highest value and not UCB. We have largely left this question for future work. As Reviewer 4 pointed out, distributional RL could be an interesting direction to take such work.
>
>
>
> **The policies of the nodes in SANE use REINFORCE but IMPALA and CLEAR use VMPO; is there a reason for this difference?** Our explicit choice was simplicity. For example, our policy is so simple (a list of logits, rather than a function approximator), the more advanced methods did not provide us any benefit. Therefore, we selected REINFORCE. Our baselines were selected later based on good public implementation. Thus the fact that the update methods differ between SANE and IMPALA/CLEAR is more incidental than an explicit decision.
>
>
>
> **In the structural updates, for node promotion the value functions of a child and its parent are compared, but for node merging, the policies are compared - is there a rationale behind these choices?:** We did consider using the policies for promotion as well, but the problem lies in thresholding. There needs to be some way to automatically determine when one node has changed “enough” from another, and value provided a natural way to do that, where policy did not. For merging, intuitively what we are aiming to do is recognize that two nodes are following the same behavior, and see if we can find some generalization in the input that encompasses both. Values being similar is insufficient information to say that we should be trying to find a common pattern.

---

> > ### Comment · AnonReviewer2 · 2020-11-24
> > **Thanks for response**
> >
> > Thank you for your replies. The results certainly have been strengthened by showing that SANE outperforms CLEAR with different replay tradeoff parameters on the MNIST experiments; the Minigrid experiments remain less convincing. Thanks also for clearing up some of the design choices for SANE. Looking forward to seeing the updated paper!

---

> ### Author Response · Authors · 2020-11-20
> **Response to Reviewer 2 (Part 2)**
>
>
> **In the MiniGrid experiments, SANE is run with either 10 or 15 top level nodes, resulting in a relatively big difference in performance for the 1st and 3rd tasks, giving some indication of the sensitivity of this hyperparameter. Why were 20 top-level nodes chosen for the MNIST experiments and was the performance sensitive to this choice?** Theoretically, the minimum number of nodes is the number of actions, if each node maps perfectly to one action. (And the nodes should be top-level for long-term stability.) MNIST has 10 actions, Minigrid has 7. Roughly doubling, we get 20 and 15. Since the more nodes we use, the longer experiments take, we sought to use the minimal necessary for the experiment. However in the more recent Minigrid experiments, we used a tree of size [20, 8], partially because we found a few optimizations to speed up training. Increasing the number of top-level nodes is not strictly beneficial; a large number means that when a new task is discovered, many (perhaps all) of them need to be “informed” via negative examples, which can slow down training. However the tradeoff is in capacity.
>
>
>
> **How about other hyperparams such as number of child nodes and the number of times a node has to have been used before it can be promoted?** The wall-clock runtime of experiments is especially sensitive to the number of children, which is unfortunate because a small number of children means constant merging, and less time for children to “distinguish” themselves, resulting in less sample efficient learning. As mentioned above we’ve moved to 8 children; hopefully as we find more performance improvements we can continue to increase this number. We experimented with the number of times a node needs to be used before being promoted, and found little sensitivity to the number. The same is true for the 3*tanh scaling of the policy, the node weight scaling when merging policies and replay buffers, and all the other constants mentioned in A.2 of the appendix, with the exception of the epsilon we use in epsilon-greedy. The number of epochs we train the nodes for is important, but unfortunately scaling this up again significantly reduces wall-clock time with diminishing gains.

---

### Official Review · AnonReviewer1 · 2020-10-27
**Interesting Idea, limited execution**

**Rating:** 5
**Confidence:** 4

**Review:**

Edit

Updated score from 4 to 5

## Summary

* The paper proposes a task agnostic CL method called SANE. It uses a hierarchical, modular tree-like architecture where nodes are neural networks (policy + critic + replay buffer).
* The architecture is quite interesting. Alongside regular parameter update operations, it also supports updates to the tree-structure.
* Each state (that a learning agent encounters) activates a new path in the tree (thus reducing negative interference) while still enabling knowledge sharing.
* Experiments on MNIST and Mini-Grid to show the usefulness of SANE over the baselines.

## Objective

* Addressing the problem of continual learning in the context of reinforcement learning.

## Strong Points

* The proposed architecture is quite intriguing.
* It has several practical desiderata like:
  * Updates on the tree-structure. More nodes can be added (on the fly) while keeping the overall complexity (of training/inference) in check.
  * Works in the task-agnostic setting, which is more general and practical than the task-aware setting.
* The writing (while missing some key details, see "Areas for Improvement") generally flows well and conveys high-level ideas.
* It is quite interesting (and motivating) that SANE retains knowledge of the previous tasks as it continues training on the new tasks. I think of the tree-structure as a way of inducing a prior for the learning agent. The tree-update operations are along the direction of learning a useful inductive bias (without hardcoding the structure). SANE could be a good step in the direction of learning priors/inductive biases from data alone.

## Areas for Improvement

* While the paper describes the high-level idea well, it is very vague in several design choices/implementation details. The lack of detail makes it hard to evaluate the usefulness of the approach. Some examples:
  * The paragraph on merging nodes is very vague. What does it mean to find the closest nodes in the policy space? Do you compute the L2 norm between the predictions of the policies of all possible inputs? How are policies combined by doing a "weighted sum of usage count"? What does "retraining once" mean. How do you decide when to stop training the merged node.

* Setup: The paper mentions that they are operating in the continual RL setup. The MNIST task is a one-step RL problem (or a one step decision-making problem). The MiniGrid setup is more exciting, but the paper considers only three tasks (difficult to evaluate the approach's utility when the number of tasks increases). It does not show convincing results on even those (more at a later point).

* Unconvincing results:

Let's start with the MiniGrid results. The paper mentions that "The Empty8x8 graph is the most insightful: CLEAR has almost perfect recall of this environment while 2-Room is training, but catastrophically forgets it during the training of Unlock.". Let us see the performance of CLEAR on Unlock -- Figure 5 (2nd row, 3rd column, MiniGrid Unlock). We see that around 1.5M steps, the CLEAR baseline reaches about 90% performance and starts to saturate. SANE is performing approximately 20% at this point, and even after training for 2.25M steps, SANE reaches around 80% success. My argument is, CLEAR starts overfitting after 1.5M steps while SANE is not overfitting even after 2.25M steps. Now, look at Figure 5 (2nd Row, 1st Column MiniGrid Empty). Around 1.5M steps, CLEAR is performing close to the SANE model.  I could argue that there is no need to train CLEAR (and other baselines) for additional 750K steps just because the SANE model needs to be updated. Updating for so many extra steps is also causing the CLEAR baseline to overfit to the Unlock task. Please note that I am not criticizing the decision to train SANE for 2.25M steps. I am criticizing the choice of training CLEAR much longer.

Regarding the MNIST results, I have the same worry. We can see that the baselines reach close to 100% very quickly by zooming into the plots, while the SANE model takes much longer. Repeating myself,  I am not criticizing the decision to train SANE much longer. I am criticizing the choice of training the other baselines much longer, thus forcing them to overfit to the current task.

* Choice of baselines: Apart from the criticism of the experiment results, I found the baselines' choice to be quite weak as well. While the authors rightly point out that task-agnostic learning is a much harder setup, they consider very few baselines (actually only one baseline, since Impala is not for CL). Regarding CLEAR and CLEARx40, I do not think CLEARx40 is a fair baseline. Using much more parameters makes the model more likely to overfit, given that the mode is always trained for a fixed number of steps, irrespective of when it starts to overfit. Some more reasonable baselines would have been:

@InProceedings{Aljundi_2019_CVPR,
author = {Aljundi, Rahaf and Kelchtermans, Klaas and Tuytelaars, Tinne},
title = {Task-Free Continual Learning},
booktitle = {Proceedings of the IEEE/CVF Conference on Computer Vision and Pattern Recognition (CVPR)},
month = {June},
year = {2019}
}

@incollection{NIPS2019_9357,
title = {Online Continual Learning with Maximal Interfered Retrieval},
author = {Aljundi, Rahaf and Belilovsky, Eugene and Tuytelaars, Tinne and Charlin, Laurent and Caccia, Massimo and Lin, Min and Page-Caccia, Lucas},
booktitle = {Advances in Neural Information Processing Systems 32},
editor = {H. Wallach and H. Larochelle and A. Beygelzimer and F. d\textquotesingle Alch\'{e}-Buc and E. Fox and R. Garnett},
pages = {11849--11860},
year = {2019},
publisher = {Curran Associates, Inc.},
url = {http://papers.nips.cc/paper/9357-online-continual-learning-with-maximal-interfered-retrieval.pdf}
}


@inproceedings{
Lee2020A,
title={A Neural Dirichlet Process Mixture Model for Task-Free Continual Learning},
author={Soochan Lee and Junsoo Ha and Dongsu Zhang and Gunhee Kim},
booktitle={International Conference on Learning Representations},
year={2020},
url={https://openreview.net/forum?id=SJxSOJStPr}
}

* Limited related work: Apart from some articles mentioned in the previous point, some recent works look at learning ensembles of "self-activating" policies.

https://icml.cc/Conferences/2020/ScheduleMultitrack?event=6293

@inproceedings{
Goyal2020Reinforcement,
title={Reinforcement Learning with Competitive  Ensembles of Information-Constrained Primitives},
author={Anirudh Goyal and Shagun Sodhani and Jonathan Binas and Xue Bin Peng and Sergey Levine and Yoshua Bengio},
booktitle={International Conference on Learning Representations},
year={2020},
url={https://openreview.net/forum?id=ryxgJTEYDr}
}




* Additional Results: It is difficult to understand what the nodes are learning. The appendix has one example, which is not sufficient to understand what the nodes are learning. It is also not clear why would the learning model try to activate different paths. This relates to the problem of learning options in RL and the commonly observed problem where only one option is active during most of the training. At the minimum, it will be useful to see some results/plots describing how the tree evolves during training and how the frequency of use of different nodes changes.

* Ablations: The paper introduces several hyper-parameters but does not consider any sort of ablations with them. This makes it difficult to understand what hyper-parameter values are important and which are set arbitrarily.

## Some other questions

Please note that these are some general questions, and I did not consider any of these design choices as "wrong" or "against the paper."

* Why did the paper use L2 norm and not KL penalty (section 2.3)

* The paper mentions that they are operating in a task-agnostic setting. While that is true, they are still training/evaluating the model in the more limited sequential (one-task-at-a-time) setting. Is there a reason to not work in the more general any-task-at-any-time setting?

* Could the authors also include a discussion of the training time for the baselines vs. SANE. My understanding is, training SANE takes a lot more time/steps.

* Could the authors add some description about how rewards are computed for the MNIST setup.

---

> ### Author Response · Authors · 2020-11-20
> **Response to Reviewer 1**
>
> Thanks so much for the extensive review, it is very helpful.
>
> **Are we overfitting CLEAR?** This is an excellent point, and one we address in one manner here, and in another manner in our response to AnonReviewer2, by tuning the replay ratio CLEAR uses. Figure 3 demonstrates early stopping CLEAR while training MNIST, comparing the result to SANE (without early stopping). We can see that CLEAR shows complete catastrophic forgetting in this case. We show only the first 3 digits for brevity; the rest of the digits follow the same pattern.
>
>
>
> The training was automatically stopped upon reaching a score of 97; since the stopping is dynamic, we chose not to average over many runs and show only one. For the remainder of the standard training period, it just ran evaluation steps for the current task (so the alignment with SANE is maintained).
>
>
>
> We also ran early stopping with the new Minigrid experiment, shown in Figure 4. These were automatically stopped upon receiving scores of 0.78, 0.95, 0.92 respectively (numbers chosen based on observation of prior plateaus). Again this only represents one run of CLEAR. We can see that in this case CLEAR beats its non-early-stopped counterpart, and is comparable to SANE except during the early stages of training Key-Corridor, where SANE performs better.
>
>
>
> It is interesting to compare MNIST, where early stopping did considerably worse, to Minigrid, where it is comparable to SANE. In the former case, good performance is achieved very quickly, so fewer samples enter the replay buffer, whereas in the latter case learning takes longer. This demonstrates that some amount of overfitting is likely occurring, and should be taken into account in our Minigrid experiments at least.
>
>
>
> **Poor choice of baselines:** Answered in General comment section.
>
>
>
> **Why L2 norm and not KL?:** We wanted the distance to be symmetric, and fast to compute. We experimented with some alternatives to L2 and found little sensitivity to the metric used. L2 norm has the benefit that we can compute the comparison of all policies to all other policies at once, as a batch, rather than needing to iterate.
>
>
>
> **Why don’t we work in the any-task-at-any-time setting?:** The any-task-at-any-time setting would work well in an environment where you have a large number of tasks and/or environments (for instance the Nethack Learning Environment), because in this case you can be sure your domain is always shifting meaningfully. In settings with fewer tasks, you end up with the problem of frequently revisiting tasks you’ve seen before, which side-steps the problem of catastrophic forgetting due to domain shift by effectively having all tasks in your domain. This does solve the problem in the few-task setting, but is not extensible to the many-task setting. Therefore we simulate the continuous domain-shift aspect of the many-task setting by only doing one task at a time, and never revisiting old tasks. (Environments such as the Nethack Learning Environment are still quite hard to solve, which is why we are using the simpler approximation in this paper.)
>
>
>
> **Training time of baselines vs SANE?** Yes, currently SANE is significantly slower: running the Minigrid experiment takes about 1 day 5 hours for SANE, and just over an hour for CLEAR.
>
>
>
> **Rewards in the MNIST setup?** Answered in the General.
>
>
>
> **Merging nodes description is vague:** Our apologies, we attempted to be as concise as possible due to the page limit. We will open source our code.
>
>
>
> Detailed Description: The policy for each node is constant across inputs (it’s not a neural network, it is simply a list of logits), which allows us to compute the L2 norm between all nodes at a layer. This is part of the advantage of the policy being so simple; it would be substantially more complex to compare function approximators. The weighted sum of usage count is given a bit more detail in the appendix, but basically we take the two usage counts (u_1 and u_2), scale them according to s = max(1, 100*tanh(u/20000)), then compute the new policy as p = (s_1 * p_1 + s_2 * p_2) / (s_1 + s_2). This is again only possible because the policy is a list of logits, not a neural net. The number of elements provided to the combined replay buffer from each node is computed in the same way. The merged critic is then trained for one epoch; in other words it is trained on all of the data in its replay buffer once. (This is a hyperparameter that can be further experimented with, 1 epoch has simply worked well for us so far.)

---

### Official Review · AnonReviewer3 · 2020-10-28

**Rating:** 4
**Confidence:** 4

**Review:**

The paper proposes SANE -- an architecture and a training algorithm for continual learning. The SANE model consists of a tree where each node can act as an RL agent and where nodes act according to the dispatching mechanism based on their reward prediction. This allows to activate and update only those agents that are specialized in the current task and thus may prevent catastrophic forgetting caused by updating the whole model.

Novelty:
Different parts of the method did have appearance in the prior literature. For example, the general idea to organize the model into a hierarhical structure with a certain kind of dispatching can be found in hierarchical softmax [1] with a more modern variant [2]. More recently, [3] proposed a similar model with a very close motivation to address catastrophic forgetting. In my opinion, authors should make a more extensive literature review and more clearly justify novelty of the method.

Clarity:
After reading the paper several times, I did not get a clear enough picture of the proposed model operates. Perhaps, a more formal algorithmic description would help.

Significance:
The empirical evaluation only consists of relatively simple experiments. MNIST, for example, is hardly representative of a real-world continual learning task. Authors do not compare SANE to string baselines that are specifically designed to prevent catastrophic forgetting, even to EWC which is cited. Even though, EWC does rely on task boundaries information, it would at the very least define the gap that exists to the methods that are agnoistic to this kind of priveledged information.
The lack of a thorough experimental study makes it difficult to argue for high significance of this work to the community.

Other comments:
 I think what authors mean by "discounted reward" and $r_{disc}$ is usually denoted as "discounted return", this should be clarified.

References:
[1] A scalable hierarchical distributed language model. Andriy Mnih and Geoffrey Hinton. 2008.
[2] Self-organized Hierarchical Softmax. Yikang Shen, Shawn Tan, Chrisopher Pal, Aaron Courville. 2017
[3] Gated Linear Networks. Joel Veness, Tor Lattimore, David Budden, Avishkar Bhoopchand, Christopher Mattern, Agnieszka Grabska-Barwinska, Eren Sezener, Jianan Wang, Peter Toth, Simon Schmitt, Marcus Hutter. 2019

---

> ### Author Response · Authors · 2020-11-20
> **Response to Reviewer 3**
>
> **How is SANE different from A scalable hierarchical distributed language model, Self-Organized Hierarchical Softmax, and Gated Linear Networks?**
>
>
>
> Thank you for the interesting related works. We will be happy to include these references and discuss them in the main paper (See discussions below). However, we want to emphasize that none of the referenced papers have been used in continual learning+RL setup (and the choices + final algorithm turns out to be significantly different). Finally, most good approaches and architectures are built on existing tools/layers/loss functions. We believe this simplicity and reliability is a virtue.
>
>
>
> Like SANE, A Scalable Hierarchical Language Model also uses a tree where a path through the tree to a particular node is utilized, to predict a probable next word (analogous to selecting a next action). However, their tree is constructed by recursively clustering, instead of by detecting and adapting to drift, a key feature of SANE. Self-Organized Hierarchical Softmax is an improvement on the same, but still doesn’t use drift detection.
>
>
>
> Gated Linear Networks is quite similar to SANE in spirit, motivated by solving catastrophic forgetting, but the mechanics of the methods are quite different; critically GLN relies on fixed, randomly sampled context functions to keep disparate tasks isolated. This differs from SANE in two key ways: first, it is not dynamic, as the contexts that activate for a particular input are always consistent, and it's just the result of that activation that changes, meaning many types of new generalizations and relationships won't be discovered. And second, it has not been demonstrated in an RL setting, where two inputs may be quite close in input space but demand different behaviors. It seems possible that the "smoothing" effect they tout (similar input means similar output) would be a hindrance in this case. Finally, a key point of their paper is that they are addressing catastrophic forgetting by avoiding backpropagation, whereas SANE utilizes it. Prior methods have certainly used gating and hierarchies before; the primary novelty of our method comes in leveraging the hierarchy to detect and adapt to drift.
>
>
>
> **Compare to better baselines?** Please see common answer for new baseline comparisons.

---

> > ### Comment · AnonReviewer3 · 2020-11-25
> > **Clarity remains an issue, EWC is not added as a baseline**
> >
> > Thank you for your reply.
> > As I understand, EWC **does** apply to the RL setting (see Section 2.2 in the paper) and thus is a valid and well-established baseline. I re-read the updated manuscript and unfortunately still don't find there a clear formal description of the method which makes me keep my current score.

---

### Official Review · AnonReviewer4 · 2020-10-28
**An interesting approach for a difficult multi-task problem**

**Rating:** 6
**Confidence:** 3

**Review:**

This work addresses multi-task learning where task boundaries are unknown. The approach is to construct a dynamic decision tree with nodes made up of small networks. Nodes are merged and promoted in the tree based on learned error bounds on value function estimates. Inference through the tree works by selecting nodes with the highest value prediction. It is an interesting approach for modular learning, even within the same environment.

The paper is clearly written. The approach is clearly explained. The empirical evaluation is thorough, with good control of the baselines.

Node promotion is done so that a new task can take a new path through the tree, avoiding catastrophic forgetting. This is done by comparing the value function estimates of a child node to its parents. A large enough difference in value function estimate triggers a node promotion. An important assumption here is that different tasks differ substantially in long term returns. This may not be the case in general. Tasks can differ in various ways apart from the total reward. Sufficient discussion is not provided on this assumption and how it affects the results.

At the end of page 5, it is claimed that sequential MNIST is “significantly more realistic”. Why is this the case?

Figure 4 is a bit hard to understand. What is the “reward” here? What is the action space for this task?

In table 1, why is the SANE accuracy for digit “0” not 100%? All other methods are 100% and this is the only class so far unless I’m misinterpreting the table.

While SANE manages to avoid catastrophic forgetting compared to the baselines, the classification accuracy for each digit does settle back down to less than 50% in most cases. Why is this the case?

How effective is the strategy of detecting task change using value function bounds in the mini world environment? The rewards do not seem to be very different. I would be interested in seeing some examples of transitions that trigger a node promotion.

The results in this environment are not so convincing. Is it perhaps because of the values being similar across tasks?

Some more discussion of how this work differs from other approaches that combine deep learning and decision tree could be included.

In section 2.3 “Training the Critics”, it is stated that the critic is trying to minimize the value v(s). Why is this done?

Future work could include using distribution RL for detecting outliers and node promotion.

This is an exciting direction of work and a difficult problem without known task boundaries. The approach is sound with decent empirical evaluations. The paper could benefit from some clarity in the explanation of the results and the assumptions made in the method. Some more in-depth analysis of the trees contracted by this method and their dynamic behavior through learning would shed some more light on some of the results.

---

> ### Author Response · Authors · 2020-11-20
> **Response to Reviewer 4**
>
> Thanks for the feedback and positive comments. We agree that distributional RL would be an interesting direction for improving our method.
>
>
>
> **With regard to the assumption that change in reward is sufficient for detecting task changes:** We do assume that any relevant changes in the environment will be reflected in the (action-conditional) discounted returns received by the agent, and we will state this more clearly in the paper. Just to clarify though: two tasks do not need to differ in their total possible return in order to be distinguished. As an example: say task A is that the agent receives reward 1 if it picks up an orange, located to its right. Task B is that the agent receives reward 1 if it picks up an apple, located to its left. After learning task A, the agent learns it will receive reward ~1 (slightly less based on discount) when it turns to the right (i.e. the selected node’s policy is to turn to the right). When placed into task B, it should learn that moving to the right will no longer get it such a high reward. Since the predicted value of the node activated during task A is no longer accurate, a new node is created, which will instead learn to turn to the left. Both tasks have the same total reward, but the action-conditional reward at each step differs, so the tasks are distinguished.
>
>
>
> If two tasks do not differ in their action-conditional returns (for instance both the apple and the orange are to the right of the agent), then the policy is inherently the same in both cases, and distinguishing between the tasks is not necessary.
>
>
>
> **Calling sequential MNIST “significantly more realistic”:** Sequential MNIST relaxes some key assumptions of existing baselines, which are more restrictive than what is found in practice in the real world. Permuted-MNIST still presents a balanced dataset during each task, in which all 10 classes are still found. Thus Sequential-MNIST relaxes the assumption that the probability of each class is consistent. Split-MNIST (5 tasks each of 2 digits) is more relaxed compared to Permuted, but still assumes that each class is presented alongside negative examples, which prevents the network from learning too naively. However, in the real world it is not necessarily the case that a new class will be presented to a learner alongside negative examples. “Significantly more realistic” is probably too strong a statement however, and we’ll change it to something more like “Sequential MNIST relaxes the assumptions other CL baselines make...".
>
>
>
> **Action space and reward for MNIST task:** Answered in the General comment we made.
>
>
>
> **Why does SANE not reach 100%:** As mentioned in the appendix, we pass the output of SANE’s policy into 3*tanh(x), to make each logit in the range (-3, 3). We found that this stabilized training; without it, after entering a new digit category, when epsilon-greedy tries the new-right digit (with very low probability) the REINFORCE gradient error is quite large, leading to instability. Using the tanh means that even after training, there will be an approximately 2.2% chance of selecting the wrong digit, even during eval (i.e. without epsilon greedy). (In future work we may stabilize in a different manner.)
>
>
>
> **Why does the accuracy of each digit settle to < 50%?:** I’m not exactly sure. When training on a new digit, the newly created node won’t have negative examples from previous digits, and thus likely starts to wrongly classify some old digits as the new digit.
>
>
>
> **How effective is node promotion for detecting task change?:** As you point out, it would be helpful to visualize when node promotions occur, to get a better sense of the answer to this question. We are working on this now.
>
>
>
> **Results are not so convincing, possibly because of similar values?:**  As mentioned above, we decided to run 3 environments that have less overlap in their goals, which demonstrates somewhat clearer behavior.
>
>
>
> **Why does “training the critics” say we minimize v(s):** This was poorly phrased and will be corrected. What the critic is doing is estimating the discounted return and the uncertainty. For v(s), it does this by minimizing the L2 loss between its estimated value and the observed return. So it’s not that v(s) is being minimized, but rather (v(s) - r_disc)^2 that is being minimized.

---

### Author Response · Authors · 2020-11-20
**General Response: Better results, baselines, and visualizations**


Thank you all so much for your constructive and positive feedback. We are pleased to report we have conducted several experiments based on reviewer suggestions, including better environments, more baselines, and trying to improve CLEAR performance with hyperparameter search (see responses to individual reviews). Our results indicate that SANE is significantly better than CLEAR on harder environments, and outperforms other baselines. We hope these new results will convince reviewers.

All figures referenced in the rebuttal have been uploaded as part of the Supplementary, in rebuttal.pdf.

 **More convincing RL results - Environment**: We present a new (better) Minigrid experiment in Figure 1. This time the three tasks have more differences from each other than in the initial set, meaning CLEAR exhibits more forgetting. The tasks are the 2 Room, then the Unlock, then an environment called Key-Corridor, where the agent needs to go through a locked door to pick up an item. We can see that SANE improves over CLEAR, undergoing less forgetting. (For comparison to early stopping, please see Figure 4, described in the response to AnonReviewer1.)

  **Better baselines?** We appreciate Reviewer 1’s suggestions of baseline, and chose to additionally run against A Neural Dirichlet Process Mixture Model, partially because of a publicly accessible official implementation. Results can be seen in Figure 2. With the method as-is, we can only compare on MNIST, as it is equipped for classification and not RL. We can see that SANE consistently beats NDPM, but partially that is due to NDPM struggling to learn the new digits. (The pattern continues through the whole set, truncated for brevity.) We used NDPM’s settings for MNIST, and did not iterate further due to time constraints. We note that while the task-Free CL looks promising for adapting to RL, but so far it has only been shown for behavior cloning, and additionally no existing open source implementation is available.

  **Better CLEAR hyperparameters?** We attempted both suggestions (early stopping and trade-off parameter). In both these cases, SANE still outperforms CLEAR (see responses below).

  **How was MNIST converted from classification to RL?** We made the selection of a class an action, with a reward of 1 if the agent gets it right and a reward of 0 if they do not. In all cases the action space is the full set of classes (size 10). (We will additionally clarify this in the paper.)

  **Visualizing node promotion:** Great suggestion. We have collected two types of data to improve visual understanding of SANE. Both were collected using the new Minigrid environment, over the first 1.3M timesteps). We can provide the full run as well as MNIST in the next couple of days. The first is a graph of when new nodes are created in the tree, shown in Figure 7. Each point represents the number of new nodes created at that timestep. With this we can see that when first starting in a new task, node creation is slow (as the new task is effectively providing negative examples to existing nodes), but when we discover that the nodes are performing worse than expected, we start to see the creation of new nodes and improved performance. The second was to generate a video of how the tree evolves over time. This video is also provided in the supplementary, as tree.mp4. When nodes are created they are pink (and appear at the bottom), and they darken to blue as they get used more (using the same scaling method as used for policy and buffer weighting).

We will be updating the final paper and code (near the deadline) after getting more feedback from reviewers to make sure we handle all the concerns including clarity issues.

---

### Author Response · Authors · 2020-11-25
**Paper Revision Submitted**

Thanks again for your thoughtful reviews. We have incorporated the feedback into the paper: improving related works, baselines, experiments, and (hopefully) clarity.

The supplementary contains updated code, longer tree creation videos, and the rebuttal.pdf with the figures we referenced in our previous rebuttal comments. (We re-ran the experiments, so the rebuttal experiments are very similar to though not identical to the results in the paper. We left the original for context.)

Due to space we were not able to fit a formal algorithmic summary. Also, as mentioned, we re-ran (almost) all of the experiments to maximally ensure consistency, but due to time (and prioritization) we have not done the increased-parameter CLEAR/IMPALA runs. We can add it (likely in the Appendix) in the camera-ready. Similarly due to time (and slow runtime), NDPM does not have the full set of runs, though we are confident in the consistency of the results, and will have the full set for the camera-ready.

---

### Decision · Program_Chairs · 2021-01-07
**Final Decision**

**Decision:**

Reject

**Comment:**

The reviewers have arrived at the consensus that this is a paper with an interesting idea, both novel and well-explained, but not quite backed up with sufficient empirical evidence. Like them, I think there is a lot of potential in modular methods for continual learning, and I know these are challenging advances to demonstrate. So I encourage you to persist, iterate and submit a stronger version of this paper in the future!